# Three-Dimensional Scanning Applied for Flexible and In Situ Calibration of Galvanometric Scanner Systems

**DOI:** 10.3390/s23042142

**Published:** 2023-02-14

**Authors:** Anders Faarbæk Mikkelstrup, Georgi Nikolaev Nikolov, Morten Kristiansen

**Affiliations:** Department of Materials and Production, Aalborg University, 9220 Aalborg, Denmark

**Keywords:** three-dimensional scanning, machine vision, automated calibration, galvanometric scanner system, neural network, laser processing

## Abstract

Galvanometric laser scanner (GLS) systems are widely used for materials processing due
to their high precision, processing velocity, and repeatability. However, GLS systems generally suffer from scan field distortions due to joint and task space relationship errors. The problem is further pronounced in robotic applications, where the GLS systems are manipulated in space, as unknown errors in the relative pose of the GLS can be introduced. This paper presents an in situ, data-driven methodology for calibrating GLS systems using 3D scanning, emphasising the flexibility, generalisation, and automated industrial integration. Three-dimensional scanning serves two primary purposes: (1) determining the relative pose between the GLS system and the calibration plate to minimise calibration errors and (2) supplying an image processing algorithm with dense and accurate data to measure the scan field distortion based on the positional deviations of marked fiducials. The measured deviations are used to train a low-complexity Radial Basis Function (RBF) network to predict and correct the distorted scan field. The proposed method shows promising results and significantly reduces the scan field distortion without the use of specialised calibration tools and with limited knowledge of the optical design of the GLS system.

## 1. Introduction

Galvanometric laser scanning (GLS) systems allow for the rapid reflection of the laser spot, with high positional repeatability and accuracy. As a result, GLS systems are common within a wide range of applications, including automotive collision avoidance [1], medical imaging [2], and materials processing [3]. This work focuses on robotic laser processing applications, i.e., remote laser cutting, forming, welding, or marking, that require an expanded workspace and more degrees of freedom than what the GLS systems offer.

Typically, GLS systems consist of one or two limited rotatable servo-driven mirrors (galvanometers) and a focusing device. Based on the application, the objective is placed before or after the scanning system, known as pre- and post-objective scanning systems. The pre-objective system utilises an f-theta objective lens to achieve a flat focal plane. The post-objective scanning system employs a dynamic focusing module (DFM) to control the focal length, such that the focus can be maintained in a limited 3D volume [4].

With either scanning system, the relationship between the mirror positions (joint space) and the Cartesian position in the workspace (task space) must be established. However, accurately mapping the joint and task space through the calibration is challenging due to the unknown geometrical, dynamical, optical, and thermal defects of the GLS system [5,6]. The influence of the individual defects varies; however, the thermal effects and the defects from mirrors and their rotary axes are commonly identified as the most significant contributors to laser beam drift [4,5].

Mounting the GLS system on an industrial manipulator adds further complexity to the calibration methodology as the relative pose between the GLS system and the calibration plane is no longer constant [7]. The overall result is undesirable positioning errors between the nominal and actual positions in the task space, even with a well-designed optical system with low manufacturing tolerances.

Robotised material processing setups are typically equipped with sensor systems for in-line inspection and monitoring. An example of such systems is the use of 3D scanning to ensure that manufactured parts comply with the required geometrical tolerances. Exploiting existing sensor setups for calibrating GLS systems not only serves a practical purpose but also allows for an in situ calibration approach. By performing measurements directly as part of the setup, external measurements that are time-consuming, challenging to automate, and error-prone are avoided [8].

The conventional approach to calibrate GLS systems is to establish the correction or look-up tables (LUT) that define the relationship between the task and joint space. Several variations of the LUT-based approach have been proposed [4,8,9]. The authors of [9] calibrated a pre-objective GLS system for laser drilling by applying Lagrange polynomials to obtain correction tables based on experimental data. The proposed method achieved a calibration accuracy with maximum deviations below 0.05 mm in a 2D setup with a limited work plane of 30 × 30 mm${}^{2}$. The authors of [8] instead proposed an in situ calibration method relying on a co-axially coupled vision system. The proposed method was validated using a pre-objective GLS system with a 100 × 100 mm${}^{2}$ work plane in a specialised experimental setup. The authors achieved a calibration accuracy of 0.014 mm at a specific reference point through six iterations. However, the applications are limited as the proposed method requires modifications to the existing optical design. The general issue with LUT-based methods is a high computational cost and sensitivity to the amount, and the properties of the experimental data [5,10].

Another approach is constructing a physical model of the GLS system [5,6,11,12]. These methods are categorised as model-driven approaches and generally rely on parameter optimisation based on accurate experimental data acquired either manually or through vision systems. In [5,11], comprehensive physical models were constructed, containing up to 26 fitting parameters that include optical, geometrical, and assembly errors. Relying on the physical model proposed by [5], the authors of [12] developed a vision-based in situ method was developed to calibrate a GLS system for additive manufacturing. The authors achieved a calibration accuracy with maximum deviations below 0.075 mm in a 350 × 350 mm${}^{2}$ work plane. It is noted that the ample parameter space often required to construct an accurate physical model leads to computationally heavy, non-convex numerical problems and sensitivity to minor changes in the optical design [13]. The numerical solution is, furthermore, commonly highly dependent on the initial value for the optimiser, requiring strong prior knowledge of the optical and mechanical design of the GLS system [13]. Limited knowledge of the optical design of some commercial GLS systems thereby presents itself as a problem.

Additionally, calibrating GLS systems directly based on traditional camera calibration methods is not possible due to the absence of a single projection centre [11]. Therefore, [14] adapted the camera model to calibrate GLS systems by introducing 12 fitting coefficients. Nevertheless, this approach suffers from similar optimisation problems and relies on the similarity between the GLS system and a camera [7]. It is possible to reduce the modelling complexity while maintaining the calibration accuracy; however, this entails the use of costly high-precision components [15].

Several data-driven methodologies have been proposed that rely on supervised statistical learning methods, such as artificial neural networks (ANN) [7,10,13]. Data-driven methods offer flexibility and depend less on the specific hardware configuration and application. Where model-driven approaches attempt to identify and eliminate sources of variance in the system by constructing a physical model representing said system, data-driven approaches instead aim to compensate for the system variance based on grey/black box models [13]. Data-driven methods have been developed to calibrate GLS systems and have proven to outperform both LUT-based and model-based approaches, both in terms of the accuracy and practicality [13]. However, the proposed data-driven methods have been used to calibrate GLS systems designed for measurement purposes, which operate under different conditions than industrial GLS systems for material processing. Studies aimed directly towards the data-driven calibration of GLS systems for material processing are limited, and the existing literature has strict requirements for the placement of the calibration plate. Moreover, none of the literature utilises the 3D scanning systems that commonly exist as a part of robotised material processing setups.

Therefore, this paper proposes a generalised and flexible in situ calibration method of GLS systems using 3D scanning, where the emphasis is placed on robotic applications. Initially, 3D scanning is applied to minimize the calibration error by determining and correcting the relative pose between the GLS system and the calibration plane. Using a simple geometrical model to map the joint and task space, fiducials can be laser marked on a calibration plate. By employing 3D scanning to supply accurate and high-density data to a set of image processing techniques, it is possible to determine the positional deviations of the marked fiducials and, thereby, the scan field distortion. Based on the measured positional deviations, a Radial Basis Function (RBF) neural network is trained to predict the necessary positional corrections across the scan field. This allows the nominal positions sent to the scanner to be corrected to compensate for the scan field distortion. The proposed methodology is developed with industrial, focusing on practicality, automation capabilities, and integration into existing systems. Therefore, the method is developed based on a generic GLS system and validated on an industrial robotic laser processing setup.

The advantages of the proposed method can be presented as follows:The data acquisition and subsequent calibration of the GLS system can be carried out directly as a part of the setup. The in situ calibration significantly reduces the calibration time and human involvement. The complete calibration and subsequent validation can be performed in less than 10 min.The use of 3D scanning to determine the relative pose between the GLS system and the calibration plate entails that only limited prior information related to the placement of the calibration plate is required, offering increased flexibility.The use of a simple RBF network allows for a trivial and efficient implementation with a low computational load into the controller of the GLS system. Additionally, the training times are low (≈6 s), allowing for rapid re-calibration.The simplified geometrical model relies on limited prior knowledge of the GLS system, underlining the generalisation ability of the proposed method. Moreover, in the case of significant changes to the optical or hardware design of the system, the geometrical model only requires minor adjustments.The in situ approach is highly suitable for industrial integration; it offers practicality and presents the achievable in-process accuracy of the system.

This paper is organised as follows: The calibration methodology is provided in Section 2, detailing the data acquisition, the geometrical model used for the task to joint space mapping, the image processing approach employed to identify marked fiducials, and the neural network for predicting and correcting the positional deviations. Section 3 presents the industrial setup on which the proposed method is validated. The resulting calibration accuracy is presented in Section 4, along with a discussion of the contributions and limitations of the proposed method. Finally, the work is concluded in Section 5, and future works are presented.

## 2. Calibration Methodology

An overview of the proposed generalised and flexible in situ calibration methodology is presented in Figure 1 and summarised below:A plain sheet metal plate serving as the calibration plate is manually placed roughly within the system’s workspace. A specialised calibration plate or reference is, hence, not required.A point cloud $G_{c}$ of the calibration plate is acquired through 3D scanning. The *c* subscript indicates that the points are given in the coordinate system of the measurement scanner. See Section 2.1.The surface normal $V_{s}$ and position $Y_{s}$ of the calibration plate are computed by fitting a plane to the acquired points $G_{c}$ representing the calibration plate. The *s* subscript indicates that the points are given in the coordinate system of the GLS system. The overall purpose is to minimise the risk of common calibration errors due to unknown deviations in the relative pose between the GLS system and the calibration plane [8]. Moreover, the need for strict positioning of the calibration plate is avoided. See Section 2.2.To visualise and measure the scan field distortion, a uniform calibration grid of *n* circular fiducials along with their Cartesian target positions $Q_{s}$ is established.If the calibration is applied, and training data exist, the trained RBF network is employed to predict the scan field deviation. The predicted deviation is then used to compute the corrected target positions $K_{s}$. Due to the low computational complexity of the RBF kernel, the corrections are performed directly on the industrial PC (IPC) controlling the GLS system. Integrating the RBF kernel on the IPC allows for a fully automatic in situ calibration, reducing calibration time and human error. See Section 2.8.The joint and task space of the GLS system are mapped utilising a simplified geometrical model. The purpose is here to establish a relationship between the control input to the scanner B and Cartesian points in the workspace of the scanner. The simple model limits the need for prior knowledge of the optical design of the GLS system and can easily be adapted to significant changes in hardware configuration. See Section 2.3.The fiducials are laser marked onto the calibration plate at a robotic pose based on the computed surface normal $V_{s}$ and position $Y_{s}$ of the calibration plate.The marked calibration plate is scanned again, following the same scanning trajectory as in Step 3. See Section 2.1.Image processing is used to locate the fiducial centres $P_{c}$ in the coordinate system of the measurement scanner to determine their actual positions. See Section 2.4.The actual positions of the marked fiducials are transformed to the coordinate system of the GLS system $P_{s}$. Thus, a direct comparison can be made to their corresponding nominal target positions $Q_{s}$. See Section 2.5.The positional deviations δ between the nominal and target positions are determined, representing the required corrections to undistort the scan field. The maximum and root mean square (RMS) of the deviations δ are computed to indicate the scan field distortion. If the RMS of the deviation δ is above a predefined tolerance $\delta_{tol}$, the calibration is run. Otherwise, the process is stopped. See Section 2.6.If the calibration is performed, the positional deviations δ are used to train a low-complexity RBF network. The trained RBF network is then used to predict the required positional corrections to undistort the scan field. The purpose of using an RBF network is to offer a simple and generalised solution with low training times. See Section 2.7.

In the below sections, the primary elements of the proposed methodology are further elaborated.

**Figure 1 sensors-23-02142-f001:**
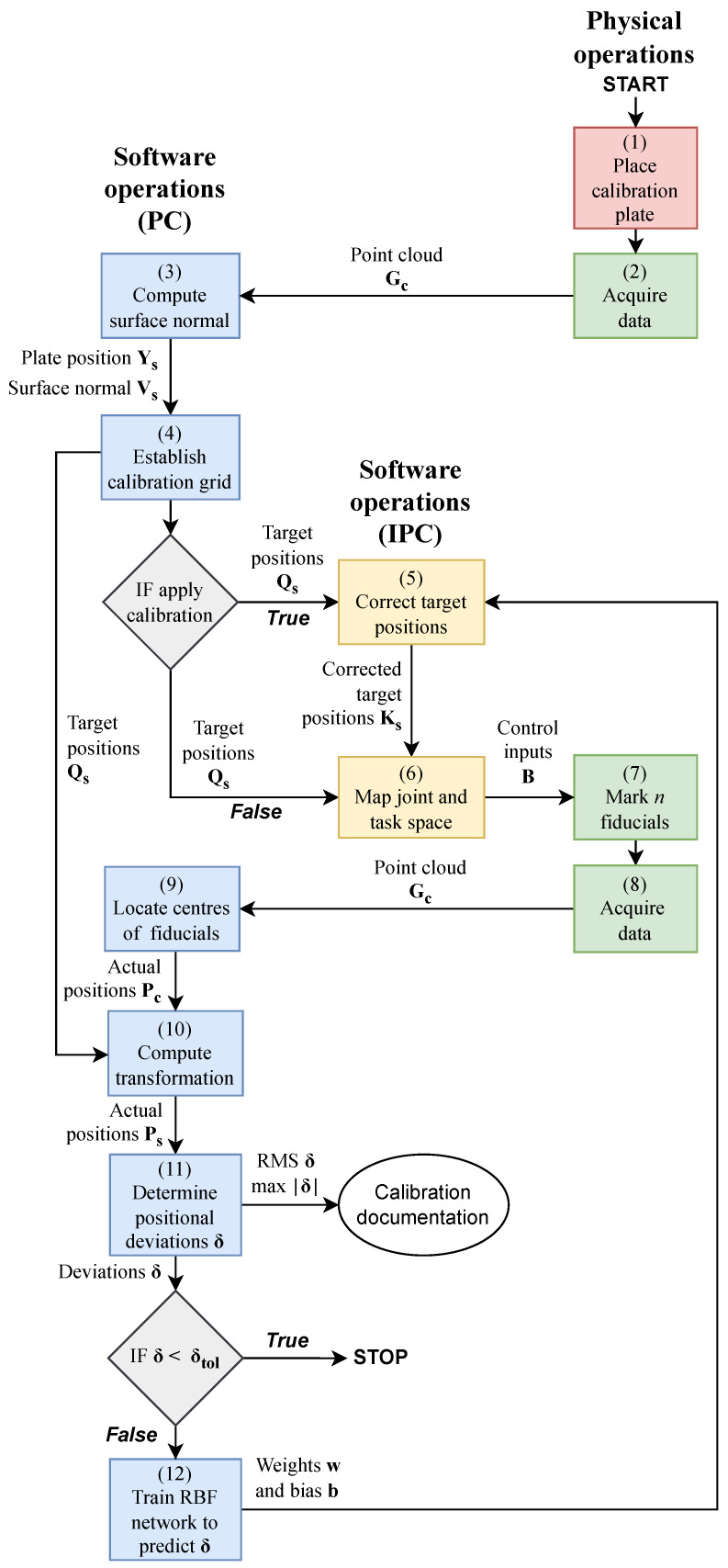
Flowchart providing an overview of the software and physical operations in the calibration methodology. The blue boxes indicate a software operation performed on the PC that handles the heavier computations. The yellow boxes indicate a software operation performed directly on the IPC controlling the GLS system. The red and green boxes represent manual and automatic physical operations.

### 2.1. Data Acquisition

Data-driven methods are known to be sensitive to the quality of the training data [7]. Thus, the data acquisition and data processing stages are essential. There are several approaches for acquiring accurate three-dimensional data, including non-contact and contact methods. As neutral networks commonly require large data sets, contact methods are considered out of scope for this paper. Non-contact-based methods rely on a range of different principles based on the application in which they are applied, i.e., stereo vision, structured light, and laser triangulation [16]. Laser line scanning based on laser triangulation is selected for its applicability in applications where rapid acquisition, high-density information, and accurate surface representation are required while maintaining a relatively low cost. Laser line scanning can further represent the reflectivity or colour of the scanned surface by measuring the amount of reflected backscatter.

As laser line scanning works by observing a projected laser profile, only 2D range data are acquired by the scanner. Therefore, the measurement scanner or object must be manipulated to achieve a full three-dimensional scan; hence, the accuracy and resolution of the acquired data depend on both the scanner and the manipulator. To achieve the highest possible robustness and accuracy of the acquired data, careful consideration of the environment, selection of scanning parameters, and planning of the scanning trajectory are necessary.

Laser speckle noise and spurious reflections are well-known challenges. A wide dynamic range of the backscattered light intensity [17,18] as well as the colour, intensity, and variation in ambient lighting can significantly affect the quality of the acquired data [19,20]. Scanning noise is especially problematic when scanning reflective surfaces such as sheet metal, presenting itself as false detections, over-saturated pixels, missing points, and a lack of accuracy.

The acquired data from the measurement scanner consists of a structured point cloud of $G_{c}=\left[x_{c},y_{c},z_{c},i_{c}\right]$ points, representing a 3D Cartesian point with an associated backscatter pixel intensity $i_{c}$ in the coordinate system of the 3D scanner $O_{c}-X_{c}Y_{c}Z_{c}$. It should be noted that the transformation between the tool centre points (TCP) of the 3D scanner and the GLS system are determined based on CAD data to align their coordinate systems.

### 2.2. Computing Surface Normal and Position of Calibration Plate

Knowledge of the relative pose between the GLS system and calibration plate is essential, as any unknown deviations in the working distance or from orthogonality will result in calibration errors. The surface normal vector $V_{s}$ of the calibration plate is determined by fitting a plane [21] to the acquired points representing the calibration plate. This permits computation of the rotation matrix R that rotates the normal vector $U_{s}=\left[0,0,1\right]$, representing the *Z*-axis of the scanner coordinate system $O_{s}-X_{s}Y_{s}Z_{s}$ onto $V_{s}$. R, hence, aligns $V_{s}$ with the *Z*-axis of the GLS system so that orthogonality can be ensured between the GLS system and the calibration plate upon marking the calibration fiducials. The working distance is corrected based on the position of the calibration plate $Y_{s}$, defined as its centre. Following the above approach, it is ensured that the calibration plate and the focal plane of the GLS system are coincident and parallel during calibration.

The same method can be further used when processing materials with the system to ensure that the relative pose between the GLS system and the working plane corresponds to the calibration.

### 2.3. Mapping the Task and Joint Space

Generally, the galvanometers of GLS systems consist of a motor, a mirror mounted on the rotational axis of the motor, and an encoder. The proposed method assumes a post-objective system in which the ingoing laser beam initially passes through the dynamic focus module (DFM). Afterwards, the laser beam continues through the objective and is then reflected by the X-mirror and, lastly, the Y-mirror. A simplified pre-objective GLS system is illustrated in Figure 2. While the DFM and the X- and Y-mirrors are in their respective zero positions, the outgoing beam hits the origin of the scanner’s coordinate system $O_{s}-X_{s}Y_{s}Z_{s}$, placed at the centre of the workspace.

As a result, manipulation of the laser beam is performed in the joint space by controlling the angles of the galvanometers and the displacement of the DFM. Nonetheless, in most practical applications, processing is performed based on Cartesian points in the task space and, therefore, the joint and task space relationship must be mapped. To limit the necessary knowledge of the optical and mechanical design of the GLS system, the mapping between the task space and joint space is based on a simplified geometrical model adapted from [4]. Resulting from the use of a simplified model, the following general assumptions apply:The mirrors are placed perpendicular to each other.The ingoing and reflected laser beam hits the mirrors on their rotational axes.The diameter of the laser beam does not exceed the mirror dimensions.

The simplified model is widely applicable as the only parameters related to the optical and mechanical design of the GLS system are the distance between the rotational axes of the X- and Y-mirrors, given as *e*, and the perpendicular distance *d* from the X-mirror to the centre of the calibration plane.

The forward kinematic equations, employed to compute a Cartesian position in the task space from a position in the joint space, are given by Equations (Equation 1) and (Equation 2).
(1)$$x_{s}=\left(d-z_{s}\right)\tan\left(\theta_{x}\right)$$
(2)$$y_{s}=\left(e+\frac{\left(d-z_{s}\right)}{\cos(\theta_{x})}\right)\tan\left(\theta_{y}\right)$$

Based on Equations (Equation 1) and (Equation 2), the inverse kinematic equations are obtained. These are presented in Equations (Equation 3)–(Equation 5). The inverse kinematic equations are used to compute the required position in the joint space to reach a specified position in the task space.
(3)$$\alpha_{x}=\frac{1}{2}\theta_{x}=\frac{1}{2}\arctan\left(\frac{x_{s}}{\left(d-z_{s}\right)}\right)$$
(4)$$\alpha_{y}=\frac{1}{2}\theta_{y}=\frac{1}{2}\arctan\left(\frac{y_{s}}{e+\sqrt{{\left(d-z_{s}\right)}^{2}+x_{s}^{2}}}\right)$$
(5)$$\Delta f_{r}=\sqrt{{\left(e+\sqrt{{\left(d-z_{s}\right)}^{2}+x_{s}^{2}}\right)}^{2}+y_{s}^{2}}-\left(e+(d-z_{s})\right)$$
where $\alpha_{x}$, $\alpha_{y}$, and $\Delta f_{r}$ define the control inputs to the scanner, representing the mechanical angles of the X- and Y-mirrors and the displacement of the DFM in relation to its zero position. A set of control inputs $B=[\alpha_{x},\alpha_{y},\Delta f_{r}]$, therefore, correspond to a nominal Cartesian position $Q_{s}=\left[x_{s},y_{s},z_{s}\right]$ in the task space, and vice versa.

It is expected that the use of a simplified geometrical model will result in some scan field distortion, as the majority of GLS systems will invalidate the above assumptions due to mechanical misalignment, laser beam drift, thermal effects, optical path errors, mount offset errors, or other unknown factors [4].

Therefore, the proposed calibration methodology, presented in the following sections, aims to compensate for the scan field distortion. The general approach relies on correcting the nominal target positions $Q_{s}$ based on the measured scan field distortion. The proposed methodology exploits that any given point $Q_{s}$ in the workspace will have a corresponding point $Q_{s}^{\prime }=\left[x_{s}^{\prime },y_{s}^{\prime },0\right]$ on the calibration plane, as illustrated in Figure 2. As such, correcting any point $Q_{s}^{\prime }$ on the focal plane will result in an adapted correction of the corresponding point $Q_{s}$ in the workspace. This permits the calibration to be performed on the focal plane while allowing for the positional corrections to be effective, even when processing outside the calibration plane.

As a result, $z_{c}$ is omitted in the following sections as all points lie on the calibration plane during calibration ($z_{c}=0$), and hence $Q_{s}^{\prime }=Q_{s}$.

### 2.4. Locating Fiducial Centres

Locating the centres of the marked circular fiducials in the acquired point cloud, illustrated in Figure 3, relies on several common image processing operations:**The 3D → 2D projection:** The point cloud is projected to 2D, such that the calibration plate is represented as a two-dimensional image $I_{c}$, i.e., $R^{3}\mapsto R^{2}$. The projection is made based on the rotation matrix R (Section 2.2) and utilising the captured backscatter intensities $i_{c}$ as the pixel intensity of the image $I_{c}$. Subsequently, a median filter is employed to $I_{c}$ to reduce noise. See Figure 4, left.**Edge detection and morphology:** The Canny edge detection operator [22] is applied to highlight the marked circles. This operation is succeeded by morphological closing and filling operations to remove noise from the binary edge image resulting from the Canny operation; Figure 4, centre.**Circle detection:** The Circular Hough Transformation (CHT) [23] is used to identify and locate circles in the processed image. See Figure 4, right. Additionally, this method offers sub-pixel accuracy, improving the detection accuracy of the circles.**The 2D → 3D projection:** Lastly, the located centre coordinates are projected back into 3D to form the matrix $P_{c}=\left[x_{c},y_{c}\right]$ of actual positions in the $O_{c}-X_{c}Y_{c}Z_{c}$ coordinate system.

Locating all existing circles is beneficial to achieve the highest calibration accuracy. Excessive scratches or contamination can lead to noise, although these have been observed to be limited on plates of rolled steel directly from the supplier.

**Figure 3 sensors-23-02142-f003:**
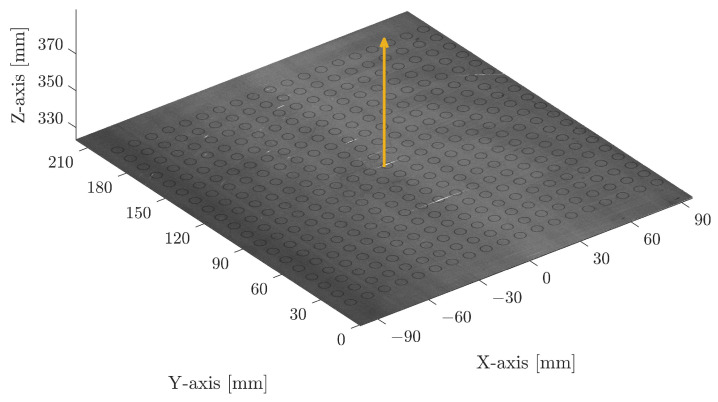
Acquired point cloud of *n* marked fiducials. The yellow arrow indicates the computed surface normal vector $V_{s}$ of the calibration plate.

**Figure 4 sensors-23-02142-f004:**
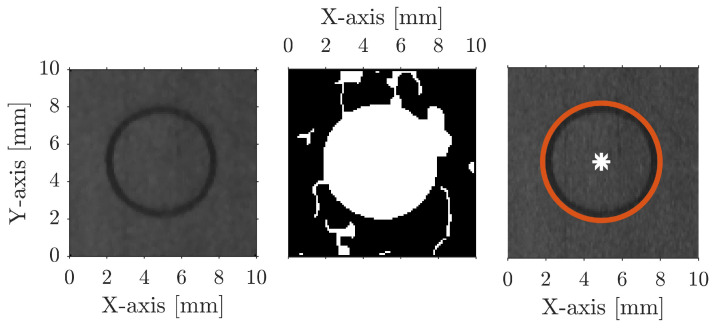
Cropped out circular fiducial of the projected point cloud, illustrating the image processing steps to locate the fiducials. A median filter is applied to the image (**left**), Canny edge detection and morphology (**centre**), and identification of circular patterns using CHT (**right**).

### 2.5. Computing the Transformation

To compare the measured actual positions $P_{c}$ with the target positions $Q_{s}$, the actual positions $P_{c}$ must be accurately transformed to the $O_{s}-X_{s}Y_{s}Z_{s}$ coordinate system of the GLS system. A rigid Euclidean transformation is used for this purpose.

Due to the optical design of the system, the distortion is limited directly on the *X*-axis and *Y*-axis, i.e., when either of the mirrors is in its respective zero position, and the other is moving. Furthermore, the distortion increases with the distance to the centre of the workspace. These properties are exploited when computing the transformation. It is assumed that the scanned calibration plate has a maximum in-plane rotation of $\pm 15^{\circ }$. An even number of points *l* are selected on each axis, ordered from smallest to largest, starting with their *X*-values and, secondly, their *Y*-values. This operation is performed for both the measured actual positions $P_{c}=\left[x_{c},y_{c}\right]$ and nominal target positions $Q_{s}=\left[x_{c},y_{c}\right]$. A set of paired points from each coordinate system is thereby created.

The paired points form the foundation to compute the Euclidean transformation, formulated as a least square optimisation problem to obtain the required rotation $R_{c}$ and translation $T_{c}$. The final rotation and translation are applied to $P_{c}=\left[x_{c},y_{c}\right]$, resulting in the actual positions $P_{s}=\left[x_{s},y_{s}\right]$ in $O_{s}-X_{s}Y_{s}Z_{s}$.

### 2.6. Determining Positional Deviations

By pairing the points of $P_{s}$ with $Q_{s}$ using the nearest neighbour approach, the positional deviations δ can then be computed as the difference between the nominal target positions $Q_{s}=\left[x_{s},y_{s}\right]$ and the actual positions $P_{s}=\left[x_{s},y_{s}\right]$, Equation (Equation 6).
(6)$$\delta =Q_{s}-P_{s}$$

The measured positional deviations δ can also serve as a means to detect drifts in the GLS system, e.g., due to wear or thermal distortions, indicating a need for calibrating or maintenance. The maximum absolute deviation and the RMS of the deviation serve as means to evaluate the accuracy of the GLS system.

However, applying the positional deviations to calibrate the system by correcting the target positions and, thereby, compensating for the measured scan field distortion presents a problem: experimentally determining the positional deviations with a high density throughout the entire scan field is impractical and time-consuming. As a result, a neural network is used to predict the positional deviations at any point across the scan field.

### 2.7. The RBF Neural Network

The Radial Basis Function (RBF) network, initially introduced by [24], can be characterised as a feed-forward neural network consisting of three layers. The RBF network is illustrated in Figure 5. RBF networks have a vast number of applications, e.g., classification [25] and function approximation [26], and are known for their simplicity and ability to generalise while maintaining a low computational complexity [27,28]. Compared to other neural networks, such as the common Multi-Layer Perceptrons (MLP) network, the RBF network further excels in increased reliability, reduced extrapolation errors, and faster convergence [29].

The RBF network is specifically selected as its characteristics make it advantageous for a robust and efficient implementation directly on the IPC that controls the GLS system. The low computational complexity is beneficial for real-time implementation, while the mathematical simplicity of the network permits native integration on IPCs, commonly limited to basic mathematical operations. Integrating the network directly on the IPC forms the basis for a fully automated calibration.

The RBF network is constructed such that it takes an input vector $U=[u_{1},u_{2}]$, here equal to a nominal position $Q_{s}=\left[x_{s},y_{s}\right]$. The output of the network $G=[g_{1},g_{2}]$ will then be equal to $\hat{\delta }=\left[{\hat{\delta }}_{x},{\hat{\delta }}_{y}\right]$, which is the predicted positional deviation along the *X*- and *Y*-axis in the given position.
(7)$$\phi_{j}\left(∥U-c_{j}∥\right)=\exp\left(-\frac{{∥U-c_{j}∥}^{2}}{2\sigma_{j}^{2}}\right)$$
(8)$$g_{k}=\sum_{j=1}^{m}w_{jk}\phi_{j}\left(U\right)+b_{k}$$

The working principle of the RBF network can be summarised as follows:(1)**Input layer:** The input vector U is fed into the network and connects the input vector to all neurons of the hidden layer.(2)**Hidden layer:** Consists of j=1…m RBF Gaussian activation functions (neurons). The similarity between the input and the stored Gaussian centre *c* is calculated based on the Euclidean norm, such that the response from the activation function increases or decreases with the distance to the centre *c*. The activation function is given by Equation (Equation 7), where $∥U-c_{i}∥$ is the Euclidean norm, while σ defines the spread of the Gaussian functions.(3)**Output layer:** Consists of k=1…2 output nodes, where each nodes outputs a linear combination of the weighted *w* responses from activation functions in the form of Equation (Equation 8). A bias term *b* is further added to each output to remove bias from the output.

As RBF networks are universal approximates, they can theoretically approximate any continuous function with arbitrary precision, assuming an appropriate selection of activation function and parameters [29]. The parameters and corresponding centres, weights, and biases are selected during the network training.

#### Training the RBF Network

The RBF network is trained using the orthogonal least-squared (OLS) method [30], following the forward centre selection approach. The OLS method was chosen due to its simplicity, speed, and deterministic behaviour, making it suitable for future integration directly on the IPC that controls the GLS system. Other approaches to determine the centres include unsupervised learning methods, such as K-means [29], which are more efficient for large and complex data sets.

By utilising the OLS method, the network is thereby iteratively constructed by selecting and adding Gaussian centres *c* from the input vector with corresponding weights *w* and bias *b*, such that the mean square error (MSE) is minimised. The network is trained using the nominal target positions $Q_{s,i}=\left[x_{s,i},y_{s,i}\right]$, i=1…n, as the input to the network along with the corresponding measured positional deviations $\delta_{i}=\left[\delta_{x,i},\delta_{y,i}\right]$ as the targets. The process is stopped when the algorithm has processed all data points or hit the specified MSE goal. As a result of the OLS method, the number of neurons *n*, centres *c*, weights *w*, and bias *b* are automatically selected. Hence, only the spread σ of the Gaussian functions and the MSE goal must be manually defined.

A small spread σ allows the network to capture local features in the data but limits its capabilities in identifying the underlying pattern. Hence, it results in poor generalisation of unseen data. Inversely, a large spread entails a close fit of the training data, allowing the network to capture global features of the data. However, overgeneralising reduces the network’s ability to identify and predict local patterns, leading to poor performance on unseen data. As such, the spread of the Gaussian functions can be used to obtain a balance between fitting the data closely and overfitting the network.

The effects of the MSE goal are similar to that of the spread. By using a stringent MSE goal, such as zero, the network learns to make predictions as close to the target values as possible. This increases the risk of overfitting the network, leading to poor generalisation abilities as the network cannot capture the underlying pattern of the training data. On the contrary, a relaxed MSE goal improves the generalisation abilities but may lead to underfitting the network. This results in poor accuracy as the network’s predictions may be far from the target values.

The selection of the hyper-parameters, here the MSE goal and spread of the Gaussian functions, is made using a grid-based search followed by a trial-and-error fine-tuning of the parameters. This approach allows for testing a wide range of parameter combinations, such that an optimal balance between the network’s generalisation performance and prediction accuracy is achieved. This enables the network to accurately predict the positional deviations across the scan field, both in and away from the training points.

An overfitted network can react highly unpredictably on unseen data, constituting a safety concern, especially when dealing with high-power laser processing. Therefore, an underfitted network is preferred to improve generalisation performance and robustness in the case of noisy data points, though at the expense of prediction accuracy.

The performance of the network during and after training is evaluated using a five-fold cross-validation approach. The selected approach permits approximation of the generalisation performance and prediction accuracy of the network without the use of new data, which in turn minimises the risk of overfitting the network [31]. However, as with any neural network, the output of the network depends on the accuracy and quality of the training data. As such, inaccurate training data will result in an inaccurate network. The uncertainty related to the applied measurement method, and hence the training data, is discussed and presented in Section 3.2.

### 2.8. Correcting the Target Positions

The trained centres, weights, and biases are fed into the RBF network, which is implemented on the IPC, controlling the GLS system. The RBF network then predicts and applies the required positional corrections to the nominal target positions $Q_{s}$. The result is the corrected target positions $K_{s}=\left[x_{k},y_{k}\right]$, given by Equation (Equation 9), that compensate for the measured scan field distortion.
(9)$$K_{s}=Q_{s}+\hat{\delta }$$

The corrected target positions are, subsequently, used to correct the control input to the scanner $B=(\theta_{x}$, $\theta_{y}$, $\Delta f_{r})$ based on the inverse kinematic equations, given by Equations (Equation 3)–(Equation 5) in Section 2.3.

## 3. Experimental Validation

The proposed method is validated based on an industrial laser processing setup, illustrated in Figure 6, along with the associated coordinate systems. The main component of the system is a customised ARGES Fibre Elephant 50 commercial GLS system that consists of dual galvanometric mirrors ($\pm 10.5^{\circ }$) and a dynamic focus module (DFM). The DFM can adjust the focal length of 490 mm ±25 mm, providing a focal plane of 300×300 mm${}^{2}$ that can be transversed at a maximum velocity of 1500 mm/s. A 3kW IPG YLS-3000 single-mode laser delivers the laser beam with a beam quality of 1.2 M${}^{2}$. The GLS system is controlled by a Beckhoff C6920-0060 IPC and mounted onto a KUKA KR 120 R2700 6-axis industrial manipulator. The relationship between the GLS system and the robot is calibrated using the accompanying cutting nozzle. The cutting nozzle is designed such that the tip of the nozzle is located at the focal point of the GLS system, permitting the use of the XYZ 4-point calibration method integrated with the robot controller. A calibration uncertainty of 0.366 mm was achieved for the robot.

The combination of hardware leads to a flexible setup with a significantly expanded workspace of the GLS system, adding additional degrees of freedom and thus permitting the processing of large-scale 3D-dimensional parts. The measurement data are acquired by a Wenglor MLWL 153 2D laser line scanner working at 405 nm. The measurement scanner is mounted in parallel with the GLS system, such that the $Z_{s}$-axis of the GLS system and the $Z_{c}$-axis of the measurement scanner are parallel. The measurement scanner offers a measuring range in $Z_{c}$ from 215 to 475 mm, resulting in a $Z_{c}$-resolution of 0.0096–0.0220 mm. The equivalent $X_{c}$-range is 150–230 mm in $X_{c}$ with a resolution between 0.079 and 0.120 mm, depending on the distance to the measured object. The linearity deviation of the line scanner is 65 μm.

The validation is performed by following the method illustrated in the flowchart, Figure 1, from Steps 1 to 12, and by substituting the target positions $Q_{s}$ with the corrected target positions $K_{s}$ in Step 6. Further details are presented in the below sections.

### 3.1. Laser Marking of Fiducials

The training data and subsequent validation for the RBF network are based on a single uniform grid marked onto a standard 1.5 mm thick AISI 316 stainless steel plate without visible surface defects. The marked steel plate thereby acts as a calibration plate. The calibration plate is clamped between two 2 mm steel plates to minimise the distortion. The upper clamping plate has a 220×220 mm${}^{2}$ cutout to expose the calibration plate. Figure 6 illustrates the fixture and calibration plate.

The marked grid consists of n=361 uniformly spaced fiducials, each composed of two concentric circles with radii $r_{1}=2.5$ mm and $r_{2}=2.7$ mm, distributed over a calibration plane of 190×190 mm${}^{2}$. The fiducials are marked at a scanning velocity of 200 mm/s with 40 W laser power, orthogonal to the calibration plate. This equates to a processing time of 89 s. The calibration plane is purposely reduced compared to the theoretical maximum focal plane to allow for localised focal adjustments in future applications (±5 mm).

### 3.2. Data Acquisition and Measurement Uncertainty

The data acquisition is executed at a linear robotic trajectory with a constant orientation and velocity of 15 mm/s. To further maximise the data quality, scans are performed at a minimal allowable working distance of roughly 325 mm, orthogonal to the scanned object, and in the absence of ambient light. The data acquisition results in a structured and uniform point cloud with X- and Y-resolution of 0.1 mm. The scanning trajectory is nominally identical in both the calibration and subsequent validation phases to minimise errors in the 3D scans by reducing deviations in the absolute positioning of the robot.

The scan field deviation measurements in the pre-calibration state are repeated five times, using identical parameters and scanning trajectories. The purpose is to validate the repeatability of the measurements. The mean RMS of the deviation ∥δ∥ (pre-calibration) across the five measurements is evaluated to 0.0819 mm with a standard deviation of 0.0028 mm, thus indicating acceptable repeatability and robustness.

The measurement accuracy and uncertainty are evaluated based on a certified GOM CP 20/MV 175×140 mm${}^{2}$ ceramic calibration plate, placed at various positions to cover the full measurement area. The calibration plate consists of 23×19 circles of r=2.3 distributed over a uniform pattern. By applying Steps 7–11 of Figure 1, the positional deviations between the nominal and measured positions of the GOM plate fiducials can be evaluated. The resulting positional deviation represents the measurement accuracy and uncertainty of the proposed method and setup. The systematic error and associated standard uncertainty (coverage factor k=1) in the X- and Y-direction is computed to 0.000±0.0914 mm and 0.000±0.0862 mm. Note that the evaluated uncertainty is roughly equivalent to the pixel size of 0.1×0.1 mm${}^{2}$.

The evaluated accuracy suggests insignificant bias in the measurements, likely due to the transformation in Step 10, Figure 1. The standard uncertainty of 0.0862 mm is expected, as the data acquisition is performed at a considerable distance from the surface, at ≈325 mm, and using a large industrial manipulator. However, the low standard uncertainty and the insignificant bias in the measurements also indicate that the robot trajectory is accurate and moves in a straight trajectory as intended. As 3D optical scanning is still a relatively new technology, it is affected by several potential error sources. These include resolution and scanning orientation, which complicate the evaluation of the measurement uncertainty [32].

### 3.3. Training the RBF Network

For training the RBF network, the MSE goal is set to 0.0005 while the spread is σ=35. The selected parameters are observed to provide a suitable balance between prediction accuracy and generalisation performance. The result is a network of m=67 neurons in the hidden layer. The achieved mean RMS prediction error of the five cross-validation rounds is 0.047 mm with a 0.0021 mm standard deviation. Hence, a calibration accuracy below the achieved RMSE of the trained network should not be expected. The low standard deviation across the validation rounds suggests appropriate stability in the chosen hyper-parameters, while the low RMSE suggests a satisfactory modelling fit. An expanded training data set could improve the prediction accuracy, however, at the expense of an overall increased calibration time, i.e., computational and experimental.

After cross-validation, the network is trained on the complete data set of n=361 points with a mean training time of 5.88 s. A single iteration of the RBF network has a mean computation time of 0.000055 s over 1000 iterations, making the RBF network suitable for real-time implementation. The computational time for the complete calibration, including data and image processing and network training, is ≈62 s.

All computational times are measured in MATLAB 2020b with an Intel Core i7 (i7-9750H) CPU @ 2.6 GHz.

## 4. Results and Discussion

The developed method was experimentally shown on the setup presented in Section 3, following the methodology shown in Figure 1. Figure 7a illustrates the scan field deviation in the pre-calibration state. It can be observed that the deviation is non-uniform and smallest at the centre of the workspace as it increases towards the edges of the scan field. This is expected and likely due to thermal effects or assembly defects, e.g., related to the mirrors and axes or the laser source, which are commonly identified as the most significant contributors to laser beam drift [4,5]. Moreover, the deviation along the *Y*-axis is lower than the *X*-axis. This can further be observed from the plotted positional deviation distribution, Figure 8. As can be seen from Table 1 and observed in the upper right corner of Figure 7a, the maximum Euclidean deviation ∥δ∥ across the entire scan field is 2.08 mm, while the RMS of the deviation is 0.85 mm. The significant deviation is expected as the simplified geometrical model does not consider any mechanical misalignment, laser beam drift, thermal effects, optical path errors, and mount offset errors, resulting in the observed non-linear scan field distortion [4]. For any practical application, the significant deviation is unacceptable and suggests that the utilised simplified model is, standalone, unsuited for calibrating the GLS system at hand.

Calibrating the GLS system using the proposed method has significantly reduced the calibration error. This can be observed by both the post-calibration scan field distortion of Figure 7b and the positional deviation distribution, Figure 8. As presented in Table 1 and observed at the outer edge of the calibration plane, the maximum positional deviation is reduced by 87.8% to 0.25 mm, while the RMS of the deviation has shown a 91.7% reduction to 0.071 mm. From Figure 7b, a relatively homogeneous scan field distortion is observed, indicating that both the data acquisition and processing along with the RBF network generally are robust.

However, the upper left and lower right corners do show outlying increases in the deviation, which likely stem from a measurement error. Because of the 0.1 mm resolution of the 3D scanner, a single-pixel shift in either direction, e.g., due to a digitisation error in the measurement scanner, a spurious reflection, or other external influences, leads to a 0.1 mm measurement error. A 3D scanner with an improved resolution could reduce the risk of measurement errors.

Calibration errors at the edges are typical; therefore, the common practice is to minimise the usable work area compared to the theoretical working area [5]. The cumulative histogram, Figure 9, of the Euclidean deviations ∥δ∥ across the scan field illustrates that 80% of the deviations are below 0.08 mm. As can also be seen in Table 1, reducing the scan field by 15 mm around the border to 150×150 mm${}^{2}$ almost halves the maximum deviation δ to 0.135 mm.

The above results confirm that the proposed method can significantly reduce the deviations and, hence, the scan field distortion. It should be noted that the calibration is performed in situ using the same hardware and materials as when the overall laser system is used for processing. Therefore, the achieved calibration accuracy is not based on specific calibration conditions and reflects the actual processing accuracy of the GLS system. Calibration methods performed in a strictly controlled environment under particular conditions, e.g., in a laboratory setup, will naturally achieve a higher calibration accuracy.

As stated in Section 1, several authors have proposed calibration approaches for GLS systems for materials processing, e.g., additive manufacturing, offering some similarities in both the application and the optical design. However, a direct quantitative comparison of the calibration accuracy to existing methods is challenging as both the hardware and optical design significantly affect the performance of the GLS systems [33], including the workspace size and positioning accuracy and repeatability. The default calibration method proposed by the manufacturer of the utilised GLS system involves laser marking a specific geometrical pattern. Subsequently, the marked pattern is manually measured using a Vernier calliper and input into the associated calibration software to correct the scan field distortion. Further details are unknown. However, the specific GLS system utilised in this paper is a customised version and, therefore, incompatible with the manufacturer’s calibration software. As a result, bench-marking the calibration accuracy with the default calibration method is not possible.

Based on the proposed methodology and results, the superiority of the proposed method compared to the existing methods can be summarised as the following:Using the proposed methodology, it is possible to perform the calibration in situ in an industrial robotised setup with limited knowledge of the optical design of the system.The achievable calibration accuracy is not dependent on certain controlled conditions as there are no strict requirements related to the placement of the calibration plate and the technical knowledge of the operator. This indicates that the achieved accuracy reflects the actual in-process accuracy of the calibrated GLS system.The results show that a GLS system can be calibrated without relying on specialised calibration tools or complex physical models and computationally heavy optimisation approaches.The proposed methodology allows for the automated calibration and validation to be conducted in less than 10 min without relying on any time-consuming and unreliable manual measurement methods.

### 4.1. Limitations and Sources of Error

#### 4.1.1. Significant Pre-Calibration Deviation

Due to the use of a simplified geometrical model, the actual correction will deviate slightly from its target correction. Thus, the measured deviation will not be fully corrected. This limits the achievable calibration accuracy in the areas of the most considerable distortion. Performing a second iteration of the calibration could solve this issue.

#### 4.1.2. Thermal Distortion

Measurement errors can occur due to thermal distortion, which directly influences the calibration accuracy. As the amount of heat induced into the surface during the laser marking process is limited, the effects are also limited. However, the probability of thermal distortion is highest at the edges of the calibration plate. Adding active cooling to the marking process could reduce the risk of errors resulting from thermal distortion.

#### 4.1.3. Absolute Positional Errors

As the distance to the calibration plate is measured, deviations in the working distance and orientation due to the misplacement of the calibration plate are minimised. However, errors in the robot’s absolute positioning and repeatability along the *Z*-axis of the robot cannot be observed and can significantly influence the calibration accuracy due to the Abbe offset. In terms of repeatability, it should be noted that the ISO9283 repeatability of the utilised robot is within ±0.05 mm. One suggested approach is to measure the radius of the marked fiducials. Though, it is challenging to distinguish if any variation is due to a calibration error, a positioning error of the robot, or simply noise.

#### 4.1.4. Euclidean Transformation

As the Euclidean transformation aligns the measured actual positions with the nominal target positions, any rotational errors around the *Z*-axis and the zero error of the GLS system cannot be measured. On the contrary, the influence of the robot’s repeatability in the XY-plane is minimised in both the marking and measurement phases.

## 5. Conclusions

This paper presents a generalised and flexible method for the in situ calibration of GLS systems applied for robotic laser processing. The main contribution of the proposed method is a generalised, fast, and flexible in situ methodology that requires no specialised tools, limited user experience, or prior knowledge related both to the placement of the calibration plate, the relative pose of the GLS system, and its optical design. The presented method applies 3D scanning combined with image processing techniques to measure the scan field distortion of the GLS system. This information is used to train an RBF neural network to predict the necessary corrections to compensate for the measured distortion. The proposed method shows promising results, with a reduction in the maximum and Euclidean scan field deviation of, respectively, 87.8% and 91.7% to 0.25 mm and 0.14 mm in the full and reduced calibration plane (150×150 mm${}^{2}$). The achieved calibration accuracy results from a compromise on the calibration accuracy to achieve a generalised, flexible, and practical solution suitable for automation and industrial integration but reflects the actual in-process accuracy.

In future works, increasing the number of measurement points could improve the robustness and generalisation performance of the RBF network, leading to more accurate predictions. Moreover, performing a second calibration iteration could potentially improve the calibration accuracy of the RBF network, as the required positional corrections will be reduced.

## Figures and Tables

**Figure 2 sensors-23-02142-f002:**
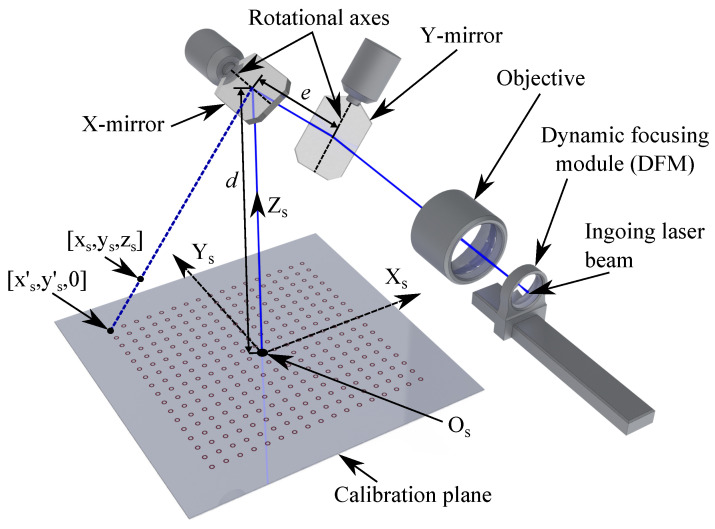
The optical chain of a pre-objective dual galvanometric system on which the geometrical model is based. The blue line indicates the path of the laser beam.

**Figure 5 sensors-23-02142-f005:**
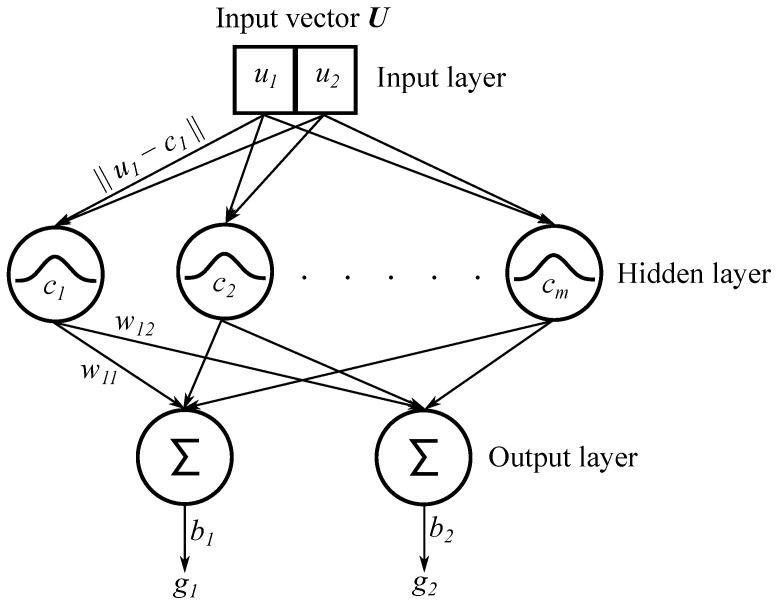
Structure of the applied RBF network.

**Figure 6 sensors-23-02142-f006:**
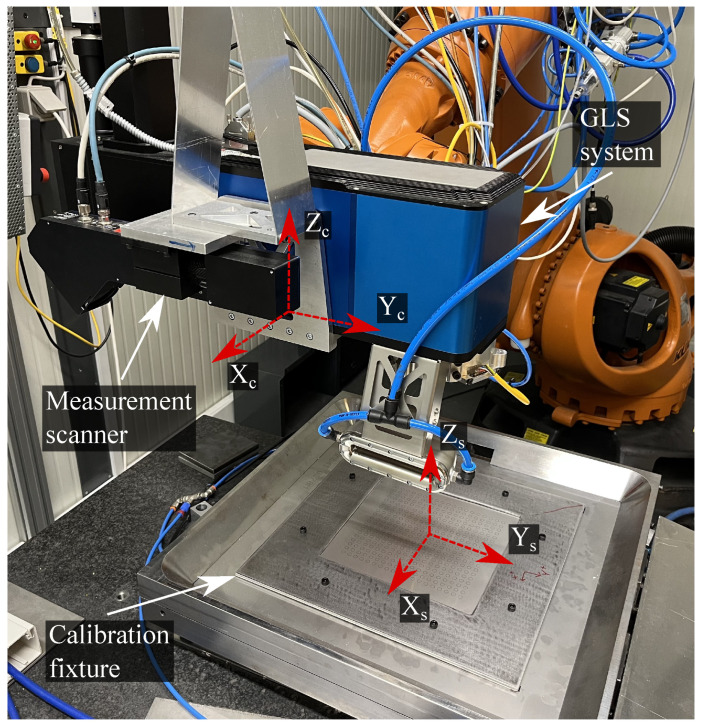
The system configuration and associated coordinate systems.

**Figure 7 sensors-23-02142-f007:**
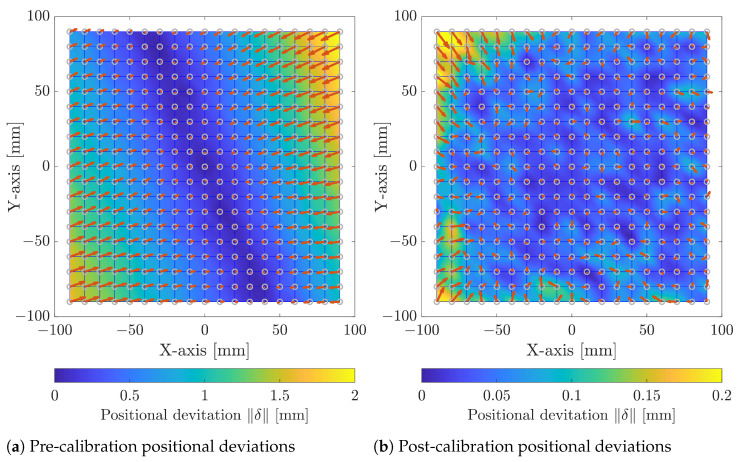
Pre- and post-calibration positional deviations δ, respectively (**a**,**b**). The grey circles indicate the target positions $P_{c}$, while the direction and scale of the red arrows indicate the direction and scale of the positional deviation. Note that the scaling of the colour bar is a factor of 10 smaller in (**b**) compared to (**a**).

**Figure 8 sensors-23-02142-f008:**
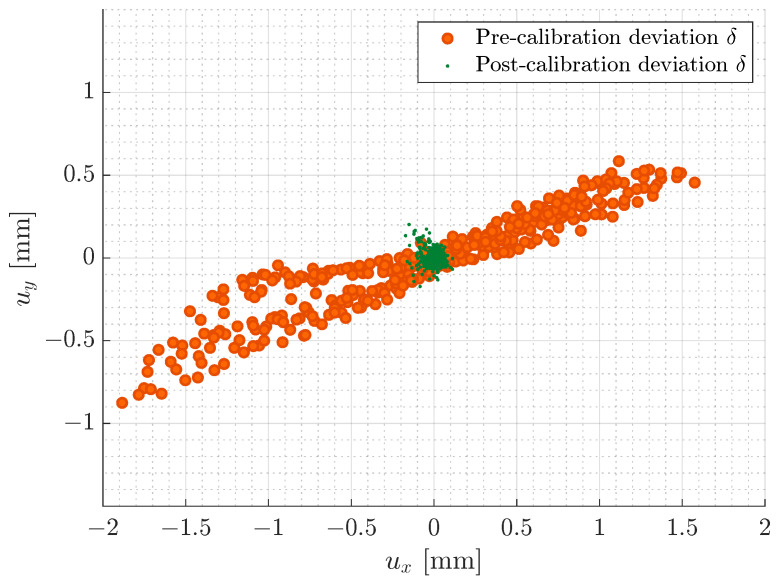
Comparison of positional deviations pre- and post-calibration.

**Figure 9 sensors-23-02142-f009:**
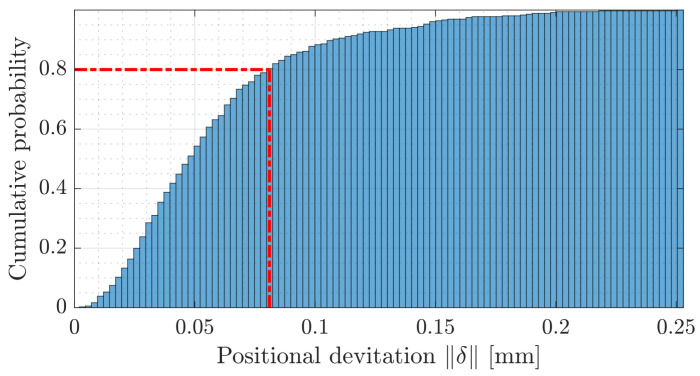
Cumulative histogram of the positional deviations δ post-calibration. As illustrated by the red line, note that 80% of the positional deviations are below 0.08 mm. The remaining 20% of the positional deviations are located at the border of the calibration area.

**Table 1 sensors-23-02142-t001:** Pre- and post-calibration results in various working areas.

	Max. [$u_{x}$, $u_{y}$, ∥δ∥]	RMS [$u_{x}$, $u_{y}$, ∥δ∥]
Pre-calibration (190 × 190 mm${}^{2}$) (mm)	[1.884, 0.876, 2.077]	[0.800, 0.301, 0.854]
Post-calibration (190 × 190 mm${}^{2}$) (mm)	[0.171, 0.202, 0.252]	[0.048, 0.051, 0.071]
Post-calibration (150 × 150 mm${}^{2}$) (mm)	[0.099, 0.099, 0.135]	[0.034, 0.034, 0.047]
Post-calibration (50 × 50 mm${}^{2}$) (mm)	[0.064, 0.058, 0.064]	[0.026, 0.023, 0.035]

## Data Availability

Data will be made available on request.

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
