# Peer review of "Three-Dimensional Scanning Applied for Flexible and In Situ Calibration of Galvanometric Scanner Systems"

_sensors, 2023, doi:10.3390/s23042142_

Round 1

Reviewer 1 Report

The present paper proposes a generalized and flexible in situ calibration of GLS systems. The paper structure is well organized, and the textual part is well written.

 I would accept the paper in the present form.

Congratulations to the authors.

Reviewer 2 Report

Your paper is significant; however, it is better if the plot is made not in Roman numerals or just made in the form of an algorithm with its syntax (Section 2).

Section 5, please distinguish between the conclusion and summary. The points in Section 5 are more suitable to be placed in the findings/novels placed in Section 1.

The geometric model in formulas 1-5 can be achieved with these five formulas. However, readers need an understanding of flow, so it is suggested that formulas 1 and 2 can be preceded by narration up to Formula 5, including the link later with formula 9.

Reviewer 3 Report

A brief summary

The authors cover the important topic of calibration methodology for Galvanometric Laser Scanning (GLS) systems using data-driven 3D scanning. The method presented focuses on flexibility, generalisation and automated industrial integration. The objectives were achieved by conducting the calibration process in situ using a machine learning method (RBF-type neural net), which significantly accelerates and increases the efficiency of the GLS calibration process.

General  comments:

1. It is not entirely clear to me how the results obtained in the calculations by the proposed method are used to calibrate the device. Is it a matter of making corrections to the measurement results or does the calibration affect the control system of the mirrors drives or the DFM module. It might be worthwhile to elaborate on this in a few sentences.

2. The authors could also consider expanding the description regarding the use of RBF networks.

2.1 Was this type of network adopted arbitrarily or did the authors compare its performance with other types of architectures?

2.2 Perhaps it would be useful to expand on the principles and the method used to set the parameters: the spread of the Gaussian function and MSE goal. How does changing them affect the outcome of the calibration process?

2.3 In line 344, the authors mention other methods of learning the network. In that case, it might be worth indicating why the OLS method was chosen and not, for example, the K-means method mentioned.

2.4 Have the authors considered the problem of overlearning the neural network and whether can it affect the effects of the calibration process? 

3. What I miss in the summary is a clear (preferably quantitative) emphasis in which the presented method is superior to other methods used as standard. 

Specific comments:

1. In line 216 after "(DMS)" there is a repetition of the word "module"

2. The text from lines 222 to 227 is repeated in lines 247 to 251

3. I do not think it is correct to start a sentence with a symbol (line 234).

4. I also do not think it is in line with the rules to use a colon to write a range of numbers in this cases (lines 387 - 389).
